# Effects of Dietary Supplementation with *Aurantiochytrium* sp. on Zebrafish Growth as Determined by Transcriptomics

**DOI:** 10.3390/ani12202794

**Published:** 2022-10-16

**Authors:** Hao Yang, Yanlin Huang, Zhiyuan Li, Yuwen Guo, Shuangfei Li, Hai Huang, Xuewei Yang, Guangli Li, Huapu Chen

**Affiliations:** 1Guangdong Research Center on Reproductive Control and Breeding Technology of Indigenous Valuable Fish Species, Guangdong Provincial Key Laboratory of Pathogenic Biology and Epidemiology for Aquatic Economic Animals, Fisheries College, Guangdong Ocean University, Zhanjiang 524088, China; 2Key Laboratory of Utilization and Conservation for Tropical Marine Bioresources of Ministry of Education, Hainan Key Laboratory for Conservation and Utilization of Tropical Marine Fishery Resources, Yazhou Bay Innovation Institute, Hainan Tropical Ocean University, Sanya 572022, China; 3Shenzhen Key Laboratory of Marine Bioresource and Eco-Environmental Science, College of Life Sciences and Oceanography, Shenzhen University, Shenzhen 518060, China; 4School of Biomedical Sciences, The Chinese University of Hong Kong, Shatin, Hong Kong 999077, China

**Keywords:** *Aurantiochytrium* sp., zebrafish, growth, transcriptome, metabolic enzymes

## Abstract

**Simple Summary:**

The high demand and price of fish oil and fishmeal, combined with the global strain on fish stocks, has attracted attention to environmentally friendly and sustainable microalgae. In this project, we fed zebrafish by feed supplementation for 56 days and found that *Aurantiochytrium* sp. extract did have a growth-promoting effect on zebrafish, with enhanced lipid metabolism and improved antioxidant capacity, reducing oxidative damage from lipid oxidation.

**Abstract:**

The marine protist *Aurantiochytrium* produces several bioactive chemicals, including EPA (eicosapentaenoic acid), DHA (docosahexaenoic acid), and other critical fish fatty acids. It has the potential to improve growth and fatty acid profiles in aquatic taxa. This study evaluated zebrafish growth performance in response to diets containing 1% to 3% *Aurantiochytrium* sp. crude extract (TE) and single extract for 56 days. Growth performance was best in the 1% TE group, and therefore, this concentration was used for further analyses of the influence of *Aurantiochytrium* sp. Levels of hepatic lipase, glucose-6-phosphate dehydrogenase, acetyl-CoA oxidase, glutathione peroxidase, and superoxide dismutase increased significantly in response to 1% TE, while malic enzyme activity, carnitine lipid acylase, acetyl-CoA carboxylase, fatty acid synthase, and malondialdehyde levels decreased. These findings suggest that *Aurantiochytrium* sp. extract can modulate lipase activity, improve lipid synthesis, and decrease oxidative damage caused by lipid peroxidation. Transcriptome analysis revealed 310 genes that were differentially expressed between the 1% TE group and the control group, including 185 up-regulated genes and 125 down-regulated genes. Kyoto Encyclopedia of Genes and Genomes (KEGG) and Gene Ontology (GO) pathway analyses of the differentially expressed genes revealed that *Aurantiochytrium* sp. extracts may influence liver metabolism, cell proliferation, motility, and signal transduction in zebrafish.

## 1. Introduction

Over the last two decades, global aquaculture production more than tripled. However, farmed fish oil production plummeted by a third, while prices more than doubled [1]. Fish oil is crucial for aquaculture because it contains critical fatty acids for fish growth, particularly n-3 highly unsaturated fatty acids (HUFA) [2]. Fish lack specific desaturases for the synthesis of n-3 and n-6 HUFA from short-chain fatty acids. They can only convert linoleic acid (LA) and linolenic acid (ALA) into 20-22C HUFA [3,4]. However, as a conventional source of unsaturated fatty acids, the extraction of n-3 HUFA from fish oil is costly and inefficient. As an alternative, marine microalgae are becoming well-known owing to their ability to synthesize long-chain unsaturated fatty acids [5].

*Aurantiochytrium* is a marine protist that produces carotenoids, squalene, sterols [6,7,8,9,10,11,12,13], and other terpenoids in addition to long-chain unsaturated fatty acids, such as eicosapentaenoic acid (EPA) and docosahexaenoic acid (DHA) [14,15,16]. *Aurantiochytrium* has a wide range of applications in aquaculture. For example, nutritional supplementation with *Aurantiochytrium* sp. increased juvenile Nile tilapia (*Oreochromis niloticus*) growth at suboptimal low temperatures (22 °C), improved the ratio of unsaturated fatty acids, and was beneficial for human consumption [17]. Guimares et al. observed that substituting *Aurantiochytrium* sp. meal for fish oil improves shrimp development and feed conversion and increases shrimp muscle DHA levels. Some studies have indicated that *Aurantiochytrium* sp. bioactive compounds can increase the quality of aquatic goods, improve meat quality, reduce organic pollutant deposition, and enhance disease resistance [18,19,20]. Most research on *Aurantiochytrium* sp. as a dietary supplement has focused on its growth-promoting and other biological effects; little is known about the mechanism by which it stimulates fish growth [17,21,22,23,24].

Unsaturated fatty acids are the primary components of *Aurantiochytrium* sp., and the liver is crucial in lipid metabolism, oxidation, and HUFA production [25]. Transcriptome sequencing technology is widely used in studies of the molecular basis of fish growth [26,27]. It can be used for comprehensive analyses of the type, structure, and expression levels of all transcripts in a tissue or cell under different conditions, revealing the molecular regulatory mechanisms underlying specific biological processes.

Therefore, zebrafish (*Danio rerio*) was utilized as a research model to assess the influence of treatment with dietary *Aurantiochytrium* sp. for 56 days on zebrafish growth performance during the critical period. The mechanism by which *Aurantiochytrium* sp. affects zebrafish growth was investigated by enzyme-linked immunosorbent assay (ELISA) and transcriptomics to detect changes in liver biochemical indicators and genes.

## 2. Materials and Methods

### 2.1. Sample Source and Diet Preparation

Shenzhen University provided the Aurantiochytrium sp. Szu445 extract and the culture and extraction methods were described previously [28]. Appendix A lists the detailed composition of the Aurantiochytrium extract (determined by GC/MS). Various amounts of Aurantiochytrium sp. Szu445 extract as well as its single extract were incorporated into the diet, established based on prior exploratory experiments: 0% *w*/*w* (Control), 1% *w*/*w* TE (T1), 2% *w*/*w* TE (T2), and 3% *w*/*w* TE (T3); 1% *w*/w DHA, 2% *w*/*w* DHA, and 3% *w*/*w* DHA; 1% *w*/*w* EPA, 2% *w*/*w* EPA, and 3% *w*/*w* EPA. In the above amounts, extracts were added to commercial feed, which was then crushed and thoroughly mixed in a pelletizer to make feed pellets. The feed pellets were spread out and dried for 72 h at 30 °C. The feed was stored at −20 °C in a refrigerator. The commercial feed contained 50% crude protein, 10% crude fat, 88% dry matter, 16% ash, and 3% crude fiber.

### 2.2. Zebrafish Treatment and Sample Collection

The China Zebrafish Resource Center supplied AB-type wild-type zebrafish, cultivated in a zebrafish culture system (Shanghai Haisheng Biological Experiment Equipment Co., Ltd., Shanghai, China). The photoperiod was maintained at 14:10, the temperature was maintained at 28 ± 0.5 °C, the pH was between 7.0 and 8.0, and the dissolved oxygen was held between 6 and 7 mg/L by the aeration system. After parental breeding, one-month-old wild-type (AB) zebrafish were randomly chosen as experimental fish. Before the experiment, the body length and weight were measured (body weight, 0.007 0.001 g; body length, 0.90 0.029 cm).

The 600 zebrafish were randomly allocated into ten groups of three tanks (10 L) each, each tank comprising 20 zebrafish. Fish were fed 5% of their body weight daily (9 am and 4 pm). Throughout the eight-week feeding experiment, the fish were weighed every two weeks to adjust the feed weight. During the 56-day rearing experiment, nine zebrafish were chosen at random every two weeks to measure growth indices, such as mean body length and mean body weight. After 56 days, fish were selected at random, anesthetized with 20 mg/L eugenols, and then euthanized on ice. The samples were snap-frozen and kept at −80 °C until RNA extraction and biochemical analysis. Guidelines for the Care and Use of Laboratory Animals were meticulously adhered to in all animal handling procedures. This protocol was approved by the Animal Research and Ethics Committee of Guangdong Ocean University (NIH Pub. No. 85-23, revised 1996).

### 2.3. Detection of Liver Biochemical Indicators

Biochemical indicators tested in this study included hepatic amylase (AMS), glucose-6-phosphate dehydrogenase (G6PD), malic enzyme (ME), pyruvate kinase (PK), acetyl-CoA oxidase (ACO), hepatic lipase (LPS), fatty acid synthase (FAS), carnitine lipid acyltransferase (CACT), acetyl-CoA (ACC), superoxide dismutase (SOD), glutathione peroxidase (GSH-Px), catalase (CAT), and malondialdehyde (MDA). The liver samples were carefully weighed, diluted tenfold with phosphate-buffered saline (PBS) (Solarbio, Beijing, China), and homogenized with a tissue crusher in an ice-water bath. After homogenization, samples were centrifuged for 10 min at 4 °C with a 3000× *g* centrifuge, and the supernatant was kept. These parameters were assayed using a microplate reader according to the manufacturer’s instructions (Multiskan MK3, Thermo Fisher Scientific, Waltham, MA, USA). Product numbers can be found in Appendix A.

### 2.4. RNA-Seq and Bioinformatics Analysis

For RNA-seq library preparation, zebrafish total RNA was extracted. Each group utilized three replicates, with each replicate including the livers of three fish. RNA was isolated using TRIzol (Invitrogen, Carlsbad, CA, USA) according to the manufacturer’s instructions. The NanoDrop 2000 was utilized to evaluate the concentration and purity of RNA (Thermo Fisher Scientific). RNA integrity was determined using the Agilent Bioanalyzer 2100 System and the RNA Nano 6000 Assay Kit (Agilent Technologies, Santa Clara, CA, USA). According to the manufacturer’s instructions, the NEBNext Ultra RNA Library Prep Kit for Il-lumina (NEB, Ipswich, MA, USA) was used to construct sequencing libraries, and index codes were added to assign sequences to each sample. The index-coded samples were clustered using a cBot Cluster Generation System equipped with a TruSeq PE Cluster Kit v4-cBot-HS (Illumina). The libraries were sequenced on the Illumina platform, and paired-end reads were generated after cluster creation. After cluster creation, libraries were sequenced on the Illumina platform. Perl scripts were used to remove adaptors and determine the Q20 and Q30 values, GC-content, and duplication rates in order to generate clean reads. The clean reads were then aligned with the genome sequence reference (GRCz11 release100). Using the reference genome as a guide, only reads containing a perfect match or one mismatch were annotated. HISAT2 was applied in order to map readings to the reference genome. Using RSEM v1.2.21, fragments per million reads per kilobase (FPKM) were utilized to assess gene expression levels [29]. Using the GOseq R package [30] and Wallenius’ non-central hypergeometric distribution, differentially expressed genes (DEGs) were analyzed for Gene Ontology (GO) enrichment. Using the Kyoto Encyclopedia of Genes and Genomes (KEGG) and KOBAS [31], an enrichment analysis of pathways was performed.

### 2.5. Real-Time Quantitative PCR (RT-qPCR) Validation

TRIzol was used to extract total RNA from liver samples according to the manufacturer’s instructions (Invitrogen). Using a Nanodrop 2000 spectrophotometer, RNA sample concentration was measured (Thermo Scientific, Waltham, MA, USA). According to the manufacturer’s instructions for the RevertAid First-strand cDNA Synthesis Kit, the same quantity of RNA was used for the reverse transcription of each sample (Thermo Scientific). LightCycler 480 (Roche, Basel, Switzerland) and SYBR Premix Ex Taq II were utilized for RT-qPCR (TaKaRa Bio Inc., Shiga, Japan). The protocol for the thermal cycler was 95 °C for 30 s, followed by 40 cycles of 95 °C for 5 s, 56 °C for 20 s, and 72 °C for 20 s. Biological triplicates and three method repetitions were performed on each sample. The reference gene β-actin was employed as an internal control to standardize mRNA levels [32]. Appendix A offers a comprehensive list of primer pairs. Using the 2^-ΔΔCt^ method, the relative gene expression levels were calculated [33].

### 2.6. Calculation of Growth Performance and Statistical Analysis

The following formulae were used to calculate growth performance:$$SR\;\left(\%\right)=N_{t}/N_{0}\times 100$$
$$RWG\;\left(\%\right)=\left(W_{t}-W_{0}\right)/\left(W_{0}\times t\right)\times 100$$
$$SGR\;\left(\%\right)=\left(\ln W_{t}-\ln W_{0}\right)/t\times 100$$
$$CF\;\left(\%\right)=W_{t}/L_{t}^{3}\times 100$$
where N_t_: final number of fish, N_0_: initial number of fish, W_t_: average fish weight after feeding (g), *W*_0_: average fish body weight at the start of feeding (g), *L_t_*: average fish length after feeding (cm), *t*: number of days of feeding (d), *SR*: survival rate, *RWG*: relative growth rate for body weight, *SGR*: specific growth rate, and *CF*: condition factor.

All data were presented as mean ± SEM (standard error of mean). IBM SPSS Statistics 24.0 was utilized for the data’s one-way ANOVA (SPSS Inc., Chicago, IL, USA). At the 95% confidence level (*p* < 0.05), the least significant difference (LSD) test was employed to determine whether or not there were significant differences between group means.

## 3. Results

### 3.1. Effect of Dietary Aurantiochytrium sp. Crude Extract and Single Extract on Growth Performance in Zebrafish

Figure 1 depicts the growth of zebrafish in various groups. Before 14 days, there were no significant differences in the changes in body weight (Figure 1A) or body length (Figure 1B) among groups of zebrafish. After 28 days, zebrafish in the *Aurantiochytrium* sp. crude extract (TE) groups showed better growth rates, in terms of both body weight and body length, than those of the single extract group and control group.

The addition of *Aurantiochytrium* sp. crude extract and its single extract to the diet had no significant influence on the CF of zebrafish (*p* > 0.05), as shown in Table 1. However, the relative weight increase rates were 20.21% and 20.27% greater in fish fed diets supplemented with 1% and 2% TE than in the control group, respectively (*p* < 0.01). The 3% TE group did not show a significant difference with the control in growth performance, which may be due to lipid peroxidation due to the high lipid content. Similar results were obtained when using a high concentration in the preliminary experiment. There were no statistically significant differences between the single extract groups with 1% EPA, 2% EPA, and 2% DHA and the control group. It is worth mentioning that 1% DHA and 3% EPA had inhibitory effects on growth. The experimental results suggest that a low concentration of *Aurantiochytrium* sp. extract has a more significant impact on zebrafish growth than that of a single extract. As a result, 1% TE was chosen for further analyses.

### 3.2. Effects of 1% Aurantiochytrium sp. Extract on Biochemical Indicators in the Zebrafish Liver

The effects of *Aurantiochytrium* sp. extract on digestive enzyme activity, energy metabolism, lipid metabolism, and antioxidant levels in the liver were evaluated. With respect to digestive enzyme activity (Figure 2A), there was no significant difference in AMS activity, and LPS activity was significantly higher in the experimental group than in the control group (*p* < 0.01). In analyses of energy metabolism (Figure 2B), G6PD levels were significantly higher in the experimental group than in the control group (*p* < 0.01), while the opposite results were obtained for ME (*p* < 0.01), and PK showed no difference between groups. In addition, ACO activity was significantly higher in the group fed 1% TE, and the activities of CACT (*p* < 0.05), ACC (*p* < 0.01), and FAS (*p* < 0.01) were higher in the control group than in the experimental group. The effects of TE on antioxidant activity (Figure 2D) were also highly significant. MDA (*p* < 0.05) and CAT levels (*p* < 0.01) were significantly higher in the control group than in the 1% TE group, while GSH-Px and SOD levels were significantly higher in the 1% TE group than in the control group (*p* < 0.01).

### 3.3. RNA-Seq of the Liver

The Illumina HiSeq high-throughput sequencing platform was used to study the zebrafish liv transcriptomes of different groups after feeding for 56 days (C, control group; T, 1% *Aurantiochytrium* sp. extract group). Following quality control, 41.60 Gb of clean data was collected, with Q30 values (i.e., defined as the proportion of sequences with a sequencing error rate of less than 0.1%) for each sample of ≥93.81%. The GC content was approximately 48.19–48.57%. A detailed summary of sequencing statistics was provided in Table 2.

The zebrafish genome was utilized as a reference to generate a sequence alignment and in subsequent analyses. Table 3 summarizes the results of the HISAT2 comparative analysis. The mapping rates to the reference genome were between 90.49% and 92.15%. The data were high quality and can be utilized to investigate DEGs further. The reads from the StringTie alignment were used for assembly and quantification.

### 3.4. Differential Gene Expression Analysis

Gene expression levels in the zebrafish liver differed significantly between the control group and the 1% *Aurantiochytrium* sp. extract group. There were 185 up-regulated genes and 125 down-regulated genes in the experimental group. Each gene was clustered according to the difference in expression (*q*-value) and fold change in expression, and a volcano plot was generated for visualization (Figure 3).

Sixteen DEGs were selected for validation by RT-qPCR (Figure 4A). Expression levels obtained by RNA-Seq and RT-qPCR were significantly correlated (*R*^2^ = 0.96088), confirming the reliability and accuracy of gene expression levels quantified by the transcriptome analysis (Figure 4C). A hierarchical clustering analysis of the screened DEGs was performed, assigning genes with the same or similar expression patterns to clusters, as shown in Figure 4B. The repeatability within groups was good, and the overall difference between groups was significant.

### 3.5. GO and KEGG Pathway Enrichment Analyses

The DEGs were subjected to a GO functional enrichment analysis. DEGs were allocated to terms within the three major functional categories of cellular component (CC), molecular function (MF), and biological process (BP).

Centrosome (GO: 0005813), extracellular space (GO: 0005576), and methionine adenosyltransferase complex (GO: 0048269) have especially substantial enrichment coefficients in CC (Figure 5A). Rho-dependent protein serine/threonine kinase activity (GO: 0072518), heme binding (GO: 0020037), and GTP-Rho binding (GO: 0017049) had the highest enrichment coefficients in MF (Figure 5B). In addition, the most significant GO secondary terms enriched by BP (Figure 5C) were circadian regulation of gene expression (GO: 0032922), cortical actin cytoskeleton organization (GO: 0030866), regulation of cell junction assembly (GO: 1901888), negative regulation of circadian rhythm (GO: 0042754), mitotic cytokinesis (GO: 0042755), and regulation of cell junction assembly (GO:1 901888) (GO: 0000281). In addition, a number of GO terms associated with cell growth and division are enriched, including mitotic DNA replication initiation (GO: 1902975), pre-replicative complex assembly involved in nuclear cell cycle DNA replication (GO: 0006267), mitochondrial cell cycle phase transition (GO: 0044772), positive regulation of cell cycle G2/M phase transition (GO: 1902751), and regulation of cyclin-dependent protein serine/threonine kinase activity (GO: 0000079).

A KEGG enrichment analysis of liver DEGs revealed 103 enriched pathways. The top 20 pathways with the most reliable enrichment were: drug metabolism—other enzymes, cell cycle, drug metabolism—cytochrome P450, metabolism of xenobiotics by cytochrome P450, pentose and glucuronate interconversions, retinol metabolism, primary bile acid biosynthesis, porphyrin, chlorophyll metabolism, vascular smooth muscle contraction, riboflavin metabolism, ABC transporters, steroid hormone biosynthesis, p53 signaling pathway, and pyrimidine metabolism.

These pathways and the typical genes involved in the pathways can be classified into substance metabolism (carbohydrate metabolism, lipid metabolism, and metabolism of cofactors and vitamins), signal transduction, and cell growth and movement. DEGs were involved in the following metabolism-related KEGG pathways: ascorbate and aldarate metabolism, pentose and glucuronate interconversions, primary bile acid biosynthesis, steroid hormone biosynthesis, glutathione metabolism, retinol metabolism, and riboflavin metabolism (Figure 6A). These functional categories will guide to reveal differences in liver metabolism in response to *Aurantiochytrium* sp. in zebrafish. Figure 6A shows the top 20 up- and down-regulated pathways (smallest q-values) and the number of enriched genes; DEGs in each KEGG pathway were annotated according to the pathway type and summarized in Figure 6B. Pathways were enriched for cellular processes, environmental information processing, and metabolism, revealing the potential effects of *Aurantiochytrium* sp. extract on metabolism, cell growth and motility, and signal transduction in the zebrafish liver.

## 4. Discussion

*Aurantiochytrium* sp. produces several biologically active compounds, including n-3 unsaturated fatty acids and carotenoids. These nutrients have the potential to influence fish physiology. The most direct indicators of fish growth and health status are growth performance measures, such as RWG and SGR. Our results reveal that a diet supplemented with 1% TE *Aurantiochytrium* sp. considerably improved RWG and SGR as well as growth performance in zebrafish over those in the control group. Similar findings have been reported in Nile tilapia (*Oreochromis niloticus*) [17], catfish (*Ictalurus punctatus*) [34], and golden pomfret (*Trachinotus ovatus*) [35].

The fish liver has high enzymatic activity, which is essential for fish growth and development. The activity of digestive enzymes influences nutrition absorption and utilization in aquatic species, which contributes to growth performance [36]. We observed a significant increase in LPS activity in the experimental group, presumably because *Aurantiochytrium* sp. is rich in n-3 polyunsaturated fatty acids, which promote lipid emulsification in the digestive tract and increase the contact area with digestive enzymes to promote lipase digestion. This was consistent with the observed enrichment for the primary bile acid biosynthetic pathway (ko00120) in the transcriptome. Uncharacterized protein LOC393433 isoform X1, 24-hydroxycholesterol 7-alpha-hydroxylase precursor, 3 beta-hydroxysteroid dehydrogenase type 7 isoform X1, and cytochrome P450 were involved in this regulatory pathway.

Glucose 6-phosphate dehydrogenase is the first enzyme in the pentose phosphate metabolic pathway; its main physiological role is to create NADPH and ribose 5-phosphate, aid in fatty acid synthesis, and maintain glutathione in the reduced form [37]. In our study, G6PD activity increased in the group with 1% *Aurantiochytrium* sp. supplementation, whereas malic enzyme (NADP-ME) levels were suppressed. ME also catalyzes the generation of NADPH, connecting the glycolytic pathway with the citric acid cycle. However, the pentose phosphate pathway is essential for the maintenance of a normal NADPH/NADP ratio, and the loss of G6PD results in high NADP levels, resulting in enhanced compensatory ME flux [38]. Starch and sucrose metabolism (ko00500), fructose and mannose metabolism (ko00051), galactose metabolism (ko00052), and pentose and glucuronic acid interconversion (ko00040) were all up-regulated to various degrees based on the transcriptome analysis. These results indicate that *Aurantiochytrium* sp. supplementation increased lipid uptake and facilitated the metabolism of differently structured glycoconjugates in zebrafish and that these pathways and processes may be associated with improved growth performance in zebrafish.

The liver, as the core site of lipid metabolism, plays a vital role in the lipid cycle [25]. FAS, CACT, and ACC all contribute to the control of lipid synthesis [39,40,41]. In our study, FAS, CACT, and ACC activity were down-regulated by Aurantiochytrium sp. ACO, the first rate-limiting enzyme in the fatty acid oxidation pathway, has the maximum activity towards medium-chain fatty acyl-CoA, and its activity diminishes with increasing chain length [42]. ACO levels were elevated in the experimental group, and related lipid cycling pathways were also enriched in the transcriptome, including fatty acid breakdown (ko00071), fatty acid biosynthesis (ko00061), and fatty acid metabolism (ko01212). These findings imply that *Aurantiochytrium* sp. may influence zebrafish growth by modulating fatty acid production and β-oxidation pathways.

Fish health is related to their antioxidant capacity. MDA is a common lipid peroxidation indicator [43]. The MDA content was dramatically reduced in the 1% *Aurantiochytrium* sp. group, showing that *Aurantiochytrium* sp. reduces lipid peroxidation in zebrafish. GSH-Px and SOD both improved to various degrees. GSH-Px functions in the removal of lipid hydroperoxides. When combined with the transcriptome analysis, our results revealed that the glutathione metabolism enzymes *gpx4a* and *mgst1.1* are involved in the control of GSH-Px. These findings are consistent with prior research showing that dietary supplementation with *Aurantiochytrium* sp. can increase the antioxidant capacity of juvenile mirror carp (*Cyprinus carpio var. specularis*) [44]. *Aurantiochytrium* carotenoids are a significant component of the natural antioxidant system. Improving the antioxidant capacity not only protects against the generation of lipid peroxidation, but also reduces the impairment of development induced by oxidative stress [45,46].

RNA-seq is a strong tool for examining the link between genotype and phenotype, providing insight into the underlying pathways and mechanisms that determine cell fate, development, and disease progression [47]. In the current investigation, zebrafish diets were supplemented with a 1% extract of *Aurantiochytrium* sp. for 56 days for RNA-seq analysis of the liver. Zebrafish DEGs were highly enriched in the GO terms catalytic activity, metabolic process, single-organism process, extracellular area, signaling, and membrane after the consumption of diets containing 1% *Aurantiochytrium* sp. extract. *Aurantiochytrium* sp. includes a variety of fatty acids, which regulate lipid metabolism and may play a role in signal transduction in zebrafish [48].

Analysis of KEGG pathways showed that DEGs were involved in environmental formation processing, genetic information processing, human disease, metabolism, and biological systems. In particular, metabolism-related pathways accounted for the majority of the top 20 enriched pathways. The pentose and glucuronide interconversion pathway, retinol metabolism pathway, primary bile acid biosynthesis pathway, porphyrin and chlorophyll metabolism pathway, ascorbate and aldose metabolism pathway, and steroid hormone biosynthesis pathway were significantly up-regulated. The differential genes of glutathione metabolism, riboflavin metabolism, and pyrimidine metabolism pathways were either up-regulated or down-regulated. They were involved in amino acid metabolism, carbohydrate metabolism, sugar biosynthesis, lipid metabolism, cofactor, and vitamin metabolism, covering most metabolic types of organic systems. In addition, some signaling and membrane transport pathways were similarly enriched, for example, p53 signaling pathway (down), ABC transporters (down), and FoxO signaling pathway (up and down). These results further suggested that the regulatory effects of *Aurantiochytrium* sp. on zebrafish growth were not mediated by a single lipid metabolic pathway.

We selected metabolic genes that may be involved in the control of zebrafish growth for qPCR validation. *mgst1.1* could reduce lipid hydroperoxides directly in membranes and plays a crucial role in hematopoietic development [49,50]. *gpx4a* could react with hydrogen peroxide and a variety of lipid hydroperoxides, including those derived from cholesterol and cholesteryl esters [51]. *odc1* is a metabolic enzyme closely related to polyamine biosynthesis associated with the initiation of apoptosis [52]. The control of glutathione metabolism suggested a role for *Aurantiochytrium* sp. extracts in zebrafish antioxidant enhancement. The liver was essential for maintaining glucose homeostasis in terms of carbohydrate metabolism. Extracts of *Aurantiochytrium* sp. modulated insulin resistance (ko04931), the insulin signaling pathway (ko04910), starch and sucrose metabolism (ko00500), and other associated pathways. In the insulin resistance pathway, protein O-GlcNAcase (*oga*), which was involved in glycoprotein metabolism and can suppress the development of type II diabetes is up-regulated [53]. In the insulin signaling pathway, protein phosphatase 1 regulatory subunit 3C-B (*ppp1r3cb*) and UDP-glucose pyrophosphorylase 2a (*ugp2a*) and ectonucleotide in starch and sucrose metabolism pyrophosphatase/phosphodiesterase family member 1 isoform X1 (*enpp1*) were also differently elevated. They contribute to glucose homeostasis and glycogen production or catabolism. Several mechanisms in lipid metabolism were also controlled. In the primary bile acid biosynthesis pathway, the cytochrome P450, family 46, subfamily A, polypeptide 1, and tandem duplicate 2 (*cyp46a1.2*) genes were increased. In the fatty acid breakdown pathway, acyl-CoA synthetase long-chain family member 1 isoform X1 (*acl1b*) was down-regulated. Within the glycerophospholipid metabolic pathway, lysophospholipid acyltransferase (*lpcat4*) was downregulated. *acl1b* catalyzes the conversion of long-chain fatty acids to acyl-coenzyme A; its downregulation reduces the oxidation of long-chain unsaturated fatty acids, hence increasing retention and deposition in muscle [54]. These data demonstrated altered expression of metabolically significant genes in the liver.

We discovered that dietary *Aurantiochytrium* sp. affects the FoxO signaling pathway, which regulates gene expression in cellular physiological events, such as apoptosis, cell cycle control, glucose metabolism, and oxidative stress resistance [49]. All these physiological events showed changes in the transcriptome. *Aurantiochytrium* sp. significantly affected genes downstream of the FoxO signaling pathway, including *sgk2b* (serine/threonine-protein kinase 2b), *gadd45ba* (growth arrest and DNA-damage-inducible, beta a), *ccnb1* (Cyclin B1), *ccnb2* (G2/mitotic-specific cyclin-B2,), *plk1* (polo-like kinase), *bnip3* (Wu:fj49c01), and cat (catalase). These genes played a role in the cell cycle, cell growth, and antioxidant defense. DHA (one of the main components of *Aurantiochytrium* sp.) negatively mediates FoxO to affect triacylglycerol metabolism in pigs [50]. However, similar results have rarely been reported in aquatic organisms. More research is needed to determine the relationship between the FoxO signaling pathway and *Aurantiochytrium* sp.

## 5. Conclusions

In summary, the liver transcriptome of zebrafish treated with dietary *Aurantiochytrium* sp. was successfully built and sequenced. Our results provide a detailed overview of the major metabolic pathways and genes that mediate the growth-promoting effects of *Aurantiochytrium* sp. *Aurantiochytrium* sp. extract has been shown to boost lipase digestion enzymes’ activity, improve lipid synthesis, and minimize oxidative damage caused by lipid peroxidation. In addition to the regulation of lipid metabolism, *Aurantiochytrium* sp. also mediates the metabolism and signal transduction of different structural sugar molecules in zebrafish. In particular, through the FoxO signaling pathway, *Aurantiochytrium* sp. may influence the metabolism of zebrafishIn. However, additional studies are needed to determine the underlying mechanism. The results of this study confirmed the growth-promoting effect of *Aurantiochytrium* sp. on zebrafish, provided insights into the mechanism by which *Aurantiochytrium* sp. regulates zebrafish growth, and provided a theoretical basis for the effective application of *Aurantiochytrium* sp. in aquaculture.

## Figures and Tables

**Figure 1 animals-12-02794-f001:**
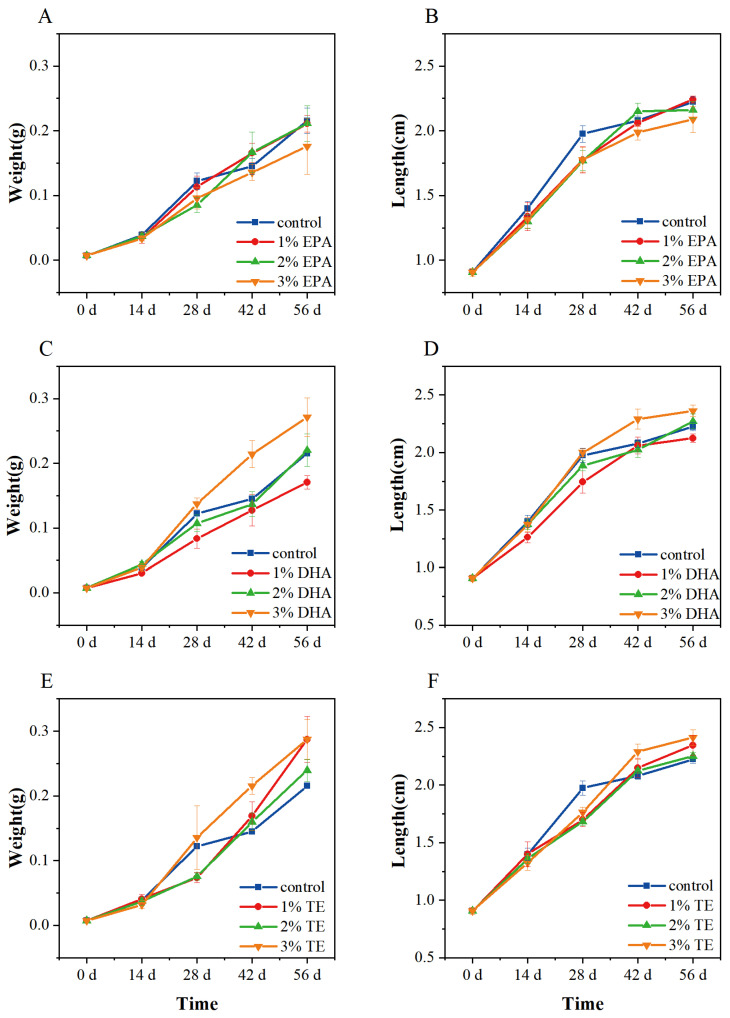
Effect of different diets on body weight and length of zebrafish. Growth data of dietary EPA group (**A**,**B**), dietary DHA group (**C**,**D**), and dietary TE group (**E**,**F**). Error bars indicate SEM (*n* = 9). EPA: eicosapentaenoic acid; DHA: docosahexaenoic acid; TE: *Aurantiochytrium* sp. extract.

**Figure 2 animals-12-02794-f002:**
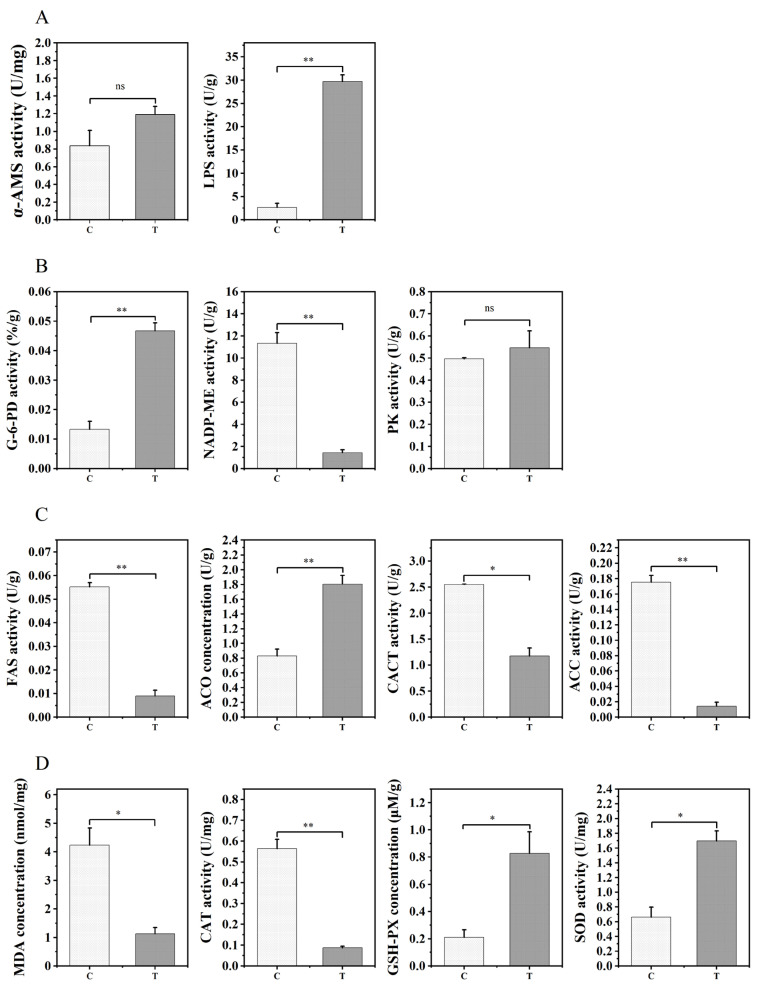
Comparative assessment of liver enzyme activities related to digestion (**A**), energy metabolism (**B**), lipid metabolism (**C**), and antioxidant defense (**D**) in the control group (C) and 1% TE group (T). Error bars indicate SEM (*n* = 9). Asterisks indicate significant differences between the two groups (* *p* < 0.05, ** *p* < 0.01), ns: not significant (*p* > 0.05) α-AMS: hepatic amylase; LPS: hepatic lipase; G-6-PD: dehydrogenase; NADP-ME: malic enzyme; PK: pyruvate kinase; FAS: fatty acid synthase; ACO: acetyl-CoA oxidase; CACT: carnitine lipid acyltransferase; ACC: acetyl-CoA; MDA: malondialdehyde; CAT: catalase; GSH–PX: glutathione peroxidase; SOD: superoxide dismutase.

**Figure 3 animals-12-02794-f003:**
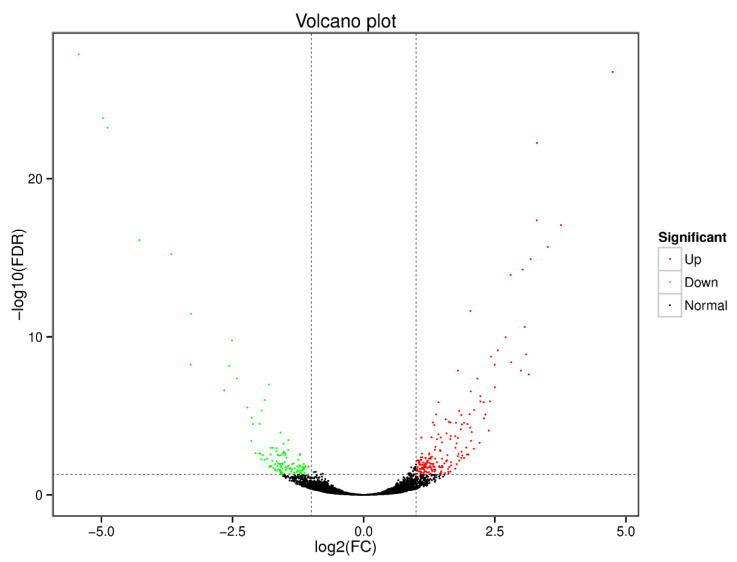
Volcano plot of differential gene expression in the zebrafish liver. Green dots represent down-regulated genes in the 1% *Aurantiochytrium* sp. extract group, red dots represent up-regulated genes, and black dots represent non-differentially expressed genes. Significance criteria were q < 0.05 and fold change ≥ 2.

**Figure 4 animals-12-02794-f004:**
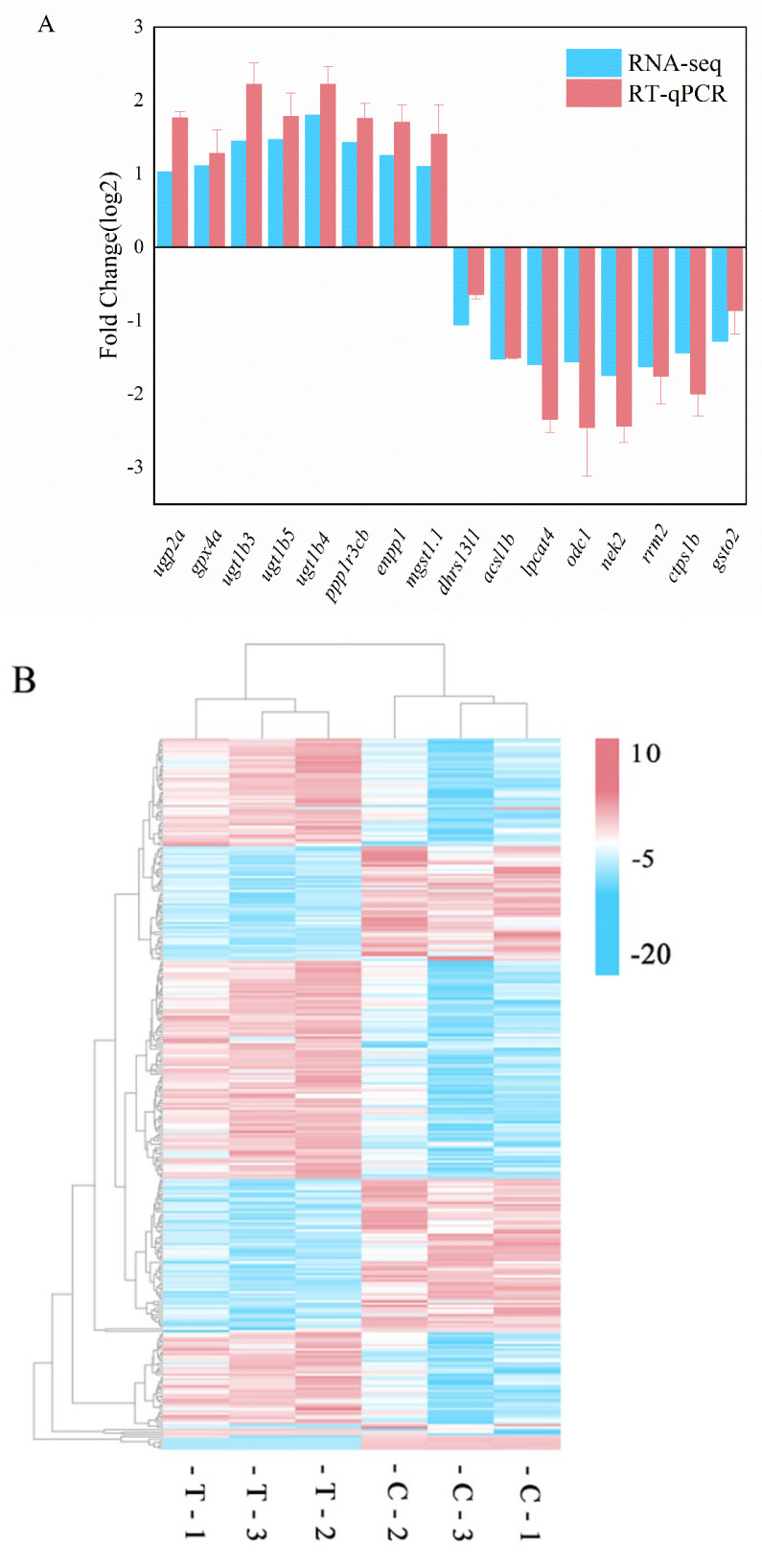
Validation of DEGs and clustering heatmap. (**A**) Validation of differential expression by RT-qPCR. Error bars indicate SEM (*n* = 3). (**B**) Cluster map of gene expression patterns demonstrating general patterns in distinct liver samples. (**C**) Correlation analysis of log2-fold change values between RNA-Seq data (*x*-axis) and RT-qPCR data (*y*-axis).

**Figure 5 animals-12-02794-f005:**
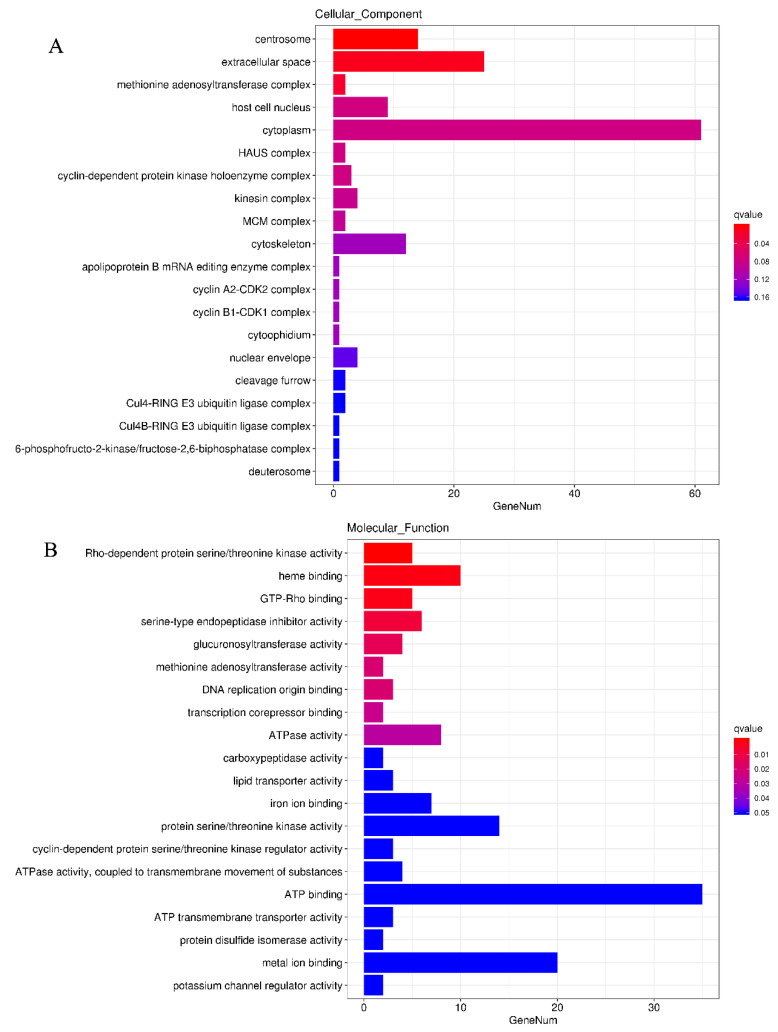
GO functional enrichment analysis of differentially expressed genes in the liver. Histograms of GO term enrichment in the (**A**) cellular component category, (**B**) molecular function category, and (**C**) biological process category.

**Figure 6 animals-12-02794-f006:**
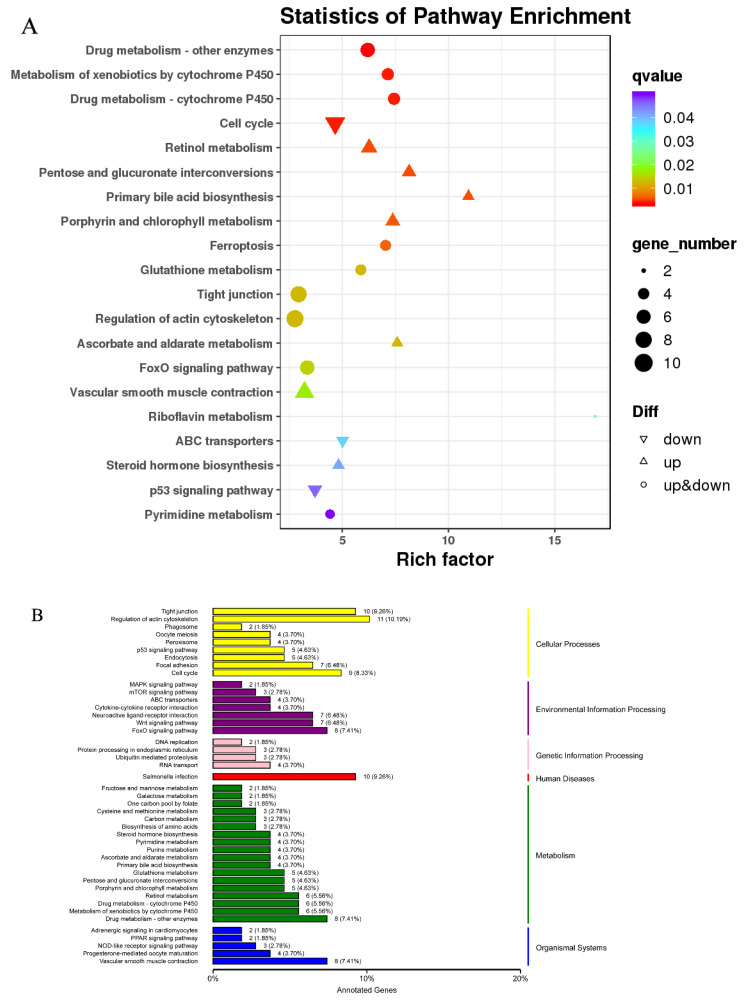
KEGG enrichment and classification. (**A**) Scatter plot of KEGG pathway enrichment results for differentially expressed genes: the ordinate represents the pathway name, the abscissa represents the enrichment factor, and the color represents the q-value (i.e., the *p*-value after correction for multiple hypothesis testing). The size of the circle indicates the number of genes involved in the pathway. (**B**) KEGG classification diagram: the left-hand vertical axis shows the KEGG pathway name, and the right-hand vertical axis shows the category corresponding to each pathway. A shared column color indicates the same category.

**Table 1 animals-12-02794-t001:** Effects of various diets on zebrafish development.

**Growth Parameters**	**Control**	**1% EPA**	**2% EPA**	**3% EPA**
RWG (%/day)	60.52 ± 4.63	59.02 ± 3.28	59.10 ± 3.35	37.81 ± 2.32 **
SGR (%/day)	6.33 ± 0.14	6.29 ± 0.10	6.30 ± 0.10	5.53 ± 0.11 **
CF (g/cm^3^)	2.07 ± 0.10	2.03 ± 0.14	2.14 ± 0.08	2.11 ± 0.13
SR (%)	93.33 ± 0.27	90.00 ± 0.47	90.00 ± 0.82	91.67 ± 0.27
**Growth Parameters**	**Control**	**1% DHA**	**2% DHA**	**3% DHA**
RWG(%/day)	60.52 ± 4.63	47.44 ± 2.99 *	61.52 ± 2.11	76.26 ± 3.12 *
SGR (%/day)	6.33 ± 0.14	5.92 ± 0.11 *	6.37 ± 0.06	6.75 ± 0.07 *
CF (g/cm^3^)	2.07 ± 0.10	2.02 ± 0.05	1.90 ± 0.14	2.19 ± 0.23
SR (%)	93.33 ± 0.27	88.33 ± 0.27	90.00 ± 0.47	93.33 ± 0.54
**Growth Parameters**	**Control**	**1% TE**	**2% TE**	**3% TE**
RWG (%/day)	60.52 ± 4.63	80.73 ± 2.33 **	80.80 ± 3.74 **	67.35 ± 4.80
SGR (%/day)	6.33 ± 0.14	6.84 ± 0.05 *	6.84 ± 0.08 *	6.52 ± 0.12
CF (g/cm^3^)	2.07 ± 0.10	2.23 ± 0.03	2.19 ± 0.09	2.00 ± 0.04
SR (%)	93.33 ± 0.27	91.67 ± 0.72	95.00 ± 0.47	93.33 ± 0.27

RWG: relative growth rate for body weight, SGR: specific growth rate, CF: condition factor, and SR: survival rate. (* *p *< 0.05, ** *p *< 0.01).

**Table 2 animals-12-02794-t002:** Sequencing data statistics.

Samples	Clean Reads	Clean Bases	GC Content	≥Q30%
C-1	20,237,339	6,043,571,206	48.47%	93.81%
C-2	22,833,729	6,803,261,102	48.57%	94.47%
C-3	29,402,492	8,774,618,350	48.43%	94.45%
T-1	22,681,184	6,757,938,492	48.19%	94.47%
T-2	22,571,833	6,730,129,406	48.49%	94.35%
T-3	21,741,708	6,489,332,016	48.26%	94.21%

**Table 3 animals-12-02794-t003:** Statistical summary of sequencing data and mapping to the zebrafish reference genome.

Sample	Total Reads	Mapped Reads	Uniq Mapped Reads	Multiple Map Reads	Reads Map to ‘+’	Reads Map to ‘−’
C-1	40,474,678	36,863,629 (91.08%)	34,099,049 (84.25%)	2,764,580 (6.83%)	18,396,027 (45.45%)	18,428,782 (45.53%)
C-2	45,667,458	42,081,877 (92.15%)	39,375,737 (86.22%)	2,706,140 (5.93%)	20,994,119 (45.97%)	21,028,413 (46.05%)
C-3	58,804,984	53,960,399 (91.76%)	50,048,173 (85.11%)	3,912,226 (6.65%)	26,914,470 (45.77%)	26,978,224 (45.88%)
T-1	45,362,368	41,432,683 (91.34%)	37,606,267 (82.90%)	3,826,416 (8.44%)	20,648,798 (45.52%)	20,715,844 (45.67%)
T-2	45,143,666	40,850,423 (90.49%)	37,875,132 (83.90%)	2,975,291 (6.59%)	20,380,508 (45.15%)	20,420,021 (45.23%)
T-3	43,483,416	39,832,673 (91.60%)	36,965,709 (85.01%)	2,866,964 (6.59%)	19,868,815 (45.69%)	19,905,921 (45.78%)

## Data Availability

The data presented in this study are available on request from the corresponding author. The data are not publicly available due to the agreement with funding bodies.

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
