# Peer review of "Effects of Dietary Supplementation with Aurantiochytrium sp. on Zebrafish Growth as Determined by Transcriptomics"

_animals, 2022, doi:10.3390/ani12202794_

Round 1
Reviewer 1 Report
Comments and suggestions to the authors
This manuscript confirms the growth-promoting effect of Aurantiochytrium sp. on zebrafish, describes the metabolic enzymes as well as metabolic pathways that Aurantiochytrium sp. regulates in the liver of zebrafish, and suggests mechanisms by which Aurantiochytrium sp. affects fish growth, which could be useful for its application in aquaculture. Moreover, the focus of the current study seems to be unique. It deserves to be published after improvement.
In the present manuscript, I found several issues. The following are my comments. I would be very happy if these comments can help improve the manuscript. After reading your manuscript, I found several problems. Here are my comments. If these comments can help improve the manuscript, I would be very pleased.
I would be very happy if they help improve the manuscript.
1. In 2.1, why did you choose two single extracts, DHA and EPA, as controls?
2. In 2.2, what is the reason for choosing 1-month-old zebrafish as the experimental subject?
3. In 2.2, the description of the experimental grouping seems to be inconsistent with the results, please check again if it is correct.
4. In 2.3, Please change " Acetyl-CoA " to " acetyl-CoA ".
5. In 2.5, why did you choose B-actin as the reference gene?
6. In 2.6, lines 158-159, you should note the expression, IBM SPSS Statistics 24.0 is used for data analysis.
7. In 3.1, " P<0.01" in P seems to need italics to be consistent with the formatting of the latter, note this type of problem.
8. In Fig.2., "*P < 0.05, **P < 0.01, ns: not significant (P > 0.05). " note the formatting issues, uniformly outside or inside parentheses.
9. In the first paragraph of 4. discussion, Latin names are not italicized.
10. Lines 377-379, the presentation seems to be problematic, please improve.
Reviewer 2 Report
This study was to estimate the effects of dietary supplementation with Aurantiochytrium on Danio rerio growth as determined by transcriptomic analysis, which is of significant meaning. However, several problems as below should be considered before publication. 1) English grammar must be improved. 2) Pay attention to the format of writing. For instance, Latin names should be italic, space between number and unit, Reference case, italics, etc. 3) Please explicitly state the nutritional formula and cultured process, especially for the culture system, the key water quality parameters, etc. Line 107-113, should give the primary determination process. Line 140-147, should provide the qPCR determination parameters, etc. P value should be italic. 4) All data must be made publically available. An accession number must be included. 5) In the line 175 and 177. In Table 1, the SR data should conclude the S.E. In line 203-204, you should check the results. Why is MDA content in accordance with the CAT activity? 6) Even with qPCR validation, the transcriptome analysis is just a prediction. Histological analysis and lipid analysis of the liver tissues are necessary. 7) What is the relationship between these selected genes? What metabolic pathways were affected by these selected genes? These questions need to be explained in the present discussion. 8) Please check the transcript data and I can't find any significant exhibited lipid transporter activity etc. in Figure 5. If so, you have to reconstruct your story. In Figure 6, authors didn't describe the specific results. 9) Please carefully consider your hypothesis.
Reviewer 3 Report
This is a significant study which deserves interest. The main results, that a diet supplemented with 1% TE Aurantiochytrium sp. considerably impfoved RWG and SGR as well as growth performance in zebrafish over those in the control group, is highly interesting. The use of zebrafish as a model provided access to much more molecular biological tools than the typical aquaculture species could do.
There are some minor points that I hope the authors would wish to consider though.
l. 26. Please avoid "loaded" expressions such as "enormous potential"
l. 171 Figure 1 has a incomplete (too short, not self-explaining) legend, and I had severe problems interpreting it! Please improve.
l. 207 The same applies to Figure 2. I was not able to interpret it.
Round 2
Reviewer 2 Report
The revised manuscript has been improved. However, there is still no sufficient discussion on the different pathways.
Author Response
Response to Reviewer 2
Question:The revised manuscript has been improved. However, there is still no sufficient discussion on the different pathways.
Answer:Thank you for your question, and it is very important to improve our manuscript for the readers. We have tried our best to enrich the discussion on the different pathways in the revised manuscript, and hope that the present discussion is suitable for publication. Thanks again.